# Large-scale analysis of the integration of enhancer-enhancer signals by promoters

Miguel Martinez-Ara[1,2], Federico Comoglio[1†], Bas van Steensel[1,2,3]*

[1]Division of Gene Regulation, Netherlands Cancer Institute, Amsterdam, Netherlands; [2]Oncode Institute, Amsterdam, Netherlands; [3]Division of Molecular Genetics, Netherlands Cancer Institute, Amsterdam, Netherlands

## eLife assessment

Understanding how genomic regulatory elements control spatiotemporal gene expression is essential for explaining cell type diversification, function, and the impact of genetic variation on disease. This **important** study provides **solid** evidence that enhancers generally combine additively to influence gene expression. Moreover, promoters, particularly weaker ones, can exhibit supra-additivity when integrating enhancer effects. These findings highlight the context-dependent nature of enhancer-promoter interactions in gene regulation, and contribute to ongoing discussions about the selectivity and combination of regulatory elements.

*For correspondence: b.v.steensel@nki.nl

Present address: †enGene Statistics GmbH, Basel, Switzerland

## Abstract

Genes are often regulated by multiple enhancers. It is poorly understood how the individual enhancer activities are combined to control promoter activity. Anecdotal evidence has shown that enhancers can combine sub-additively, additively, synergistically, or redundantly. However, it is not clear which of these modes are more frequent in mammalian genomes. Here, we systematically tested how pairs of enhancers activate promoters using a three-way combinatorial reporter assay in mouse embryonic stem cells. By assaying about 69,000 enhancer-enhancer-promoter combinations we found that enhancer pairs generally combine near-additively. This behaviour was conserved across seven developmental promoters tested. Surprisingly, these promoters scale the enhancer signals in a non-linear manner that depends on promoter strength. A housekeeping promoter showed an overall different response to enhancer pairs, and a smaller dynamic range. Thus, our data indicate that enhancers mostly act additively, but promoters transform their collective effect non-linearly.

## Introduction

### Enhancers: background

Enhancers are DNA elements, typically a few hundred base pairs long, that bind transcription factors (TFs) and boost the activity of genes in cis (*Banerji et al., 1981*). Over the past decades, many insights have been obtained into the molecular architecture of enhancers, the roles of TFs and their cofactors, the interplay of these factors with chromatin, and the diversity of mechanisms that govern the interactions of enhancers with promoters (*Kim and Wysocka, 2023*).

### MPRAs to probe enhancers

In recent years, massively parallel reporter assays (MPRAs) have been developed to systematically dissect the sequence features that determine the functionality of regulatory elements by probing thousands or even millions of DNA elements in a multiplexed manner (*Inoue and Ahituv, 2015*; *Klein et al., 2020*; *Gallego Romero and Lea, 2023*). Among others, two-way combinatorial MPRA studies

that tested a large number of enhancer-promoter pairs revealed a variable degree of intrinsic compatibility between enhancers and promoters (*Zabidi et al., 2015*; *Bergman et al., 2022*; *Martinez-Ara et al., 2022*), in line with earlier evidence from analyses of single genomic loci (*Bertolino and Singh, 2002*; *Chang et al., 2004*; *Vakoc et al., 2005*; *Jing et al., 2008*; *Deng et al., 2012*).

## Modes of enhancer interplay are still poorly understood

Surveys of mammalian genomes suggest that there are many more enhancers than genes (*The ENCODE Project Consortium, 2012*). Indeed, genes are frequently controlled by multiple enhancers (*Shlyueva et al., 2014*; *Gasperini et al., 2020*; *Kim and Wysocka, 2023*). Often, such enhancers are clustered and jointly regulate a nearby tissue-specific gene (*Tuan et al., 1985*; *Grosveld et al., 1987*; *Tolhuis et al., 2002*; *Deng et al., 2012*; *Hnisz et al., 2013*; *Whyte et al., 2013*; *Hay et al., 2016*; *Schoenfelder and Fraser, 2019*; *Brosh et al., 2023*). However, how multiple enhancers act together to control a single promoter is still poorly understood. Several studies have found that enhancers can combine either additively (*Bothma et al., 2015*; *Lam et al., 2015*; *Dukler et al., 2016*; *Hay et al., 2016*; *Will et al., 2017*; *Lin et al., 2022*), supra-additively (indicating synergy) (*Bothma et al., 2015*; *Lam et al., 2015*; *Joo et al., 2016*; *Shin et al., 2016*; *Carleton et al., 2017*; *Thomas et al., 2021*; *Lin et al., 2022*; *Brosh et al., 2023*), or sub-additively (indicating redundance) (*Moorthy et al., 2017*; *Osterwalder et al., 2018*). Nevertheless, all of these studies tested only small numbers of enhancers in one or a few genomic loci. Therefore, it is still unclear what the globally predominant mode of functional interaction is between enhancers, and how this may depend on the target promoter. For this, more scalable approaches are needed. Here, we report such an approach.

## Systematic approach

To systematically assess enhancer-enhancer (EE) interplay and the collective effect on promoter activity, we designed a three-way combinatorial MPRA based on our previous combinatorial reporters (*Martinez-Ara et al., 2022*). This assay allowed us to test tens of thousands of enhancer-enhancer-promoter (EEP) combinations in a uniform setting (*Inoue and Ahituv, 2015*; *van Arensbergen et al., 2017*; *Martinez-Ara et al., 2022*). In order to test how different promoters might affect the enhancer interplay and their output, we not only varied enhancer pairs, but also the identity of the promoter in the three-way combinations.

## Results and significance

By testing about 69,000 EEP combinations in mouse embryonic stem cells (mESCs) we found that pairs of enhancers mostly combine near-additively. Furthermore, promoter choice affects this EE interplay, and promoters integrate the enhancer effects in a non-linear manner. Our three-way combinatorial approach provides further insights into two key aspects of gene regulation: how enhancers combine their effects, and how promoters integrate enhancer signals.

# Results

## Construction of libraries/experimental design

### Triple combinations approach

To systematically test how pairs of enhancers work together and in turn activate promoters, we implemented a three-way combinatorial approach in an MPRA. We designed a cloning strategy that enabled us to construct libraries of tens of thousands of reporters that each contain a different EEP combination (*Figure 1A and B*).

### Model system and promoter selection

For these experiments we chose mESCs as a model. We selected eight promoters that are active in mESCs. Of these, seven are from tightly regulated genes that are involved in the control of the pluripotency state of the cells or their exit from this state (*Klf2, Sox2, Nanog, Otx2, Lefty1, Ffg5, Tbx3*) (*Acampora et al., 2013*; *Dunn et al., 2014*; *Kim et al., 2014*; *Thomas et al., 2021*). In addition, we included the promoter of the housekeeping (*Hounkpe et al., 2021*) gene *Ap1m1*, which neighbours the *Klf2* gene. From each promoter we approximately included the −350 to +50 bp region around

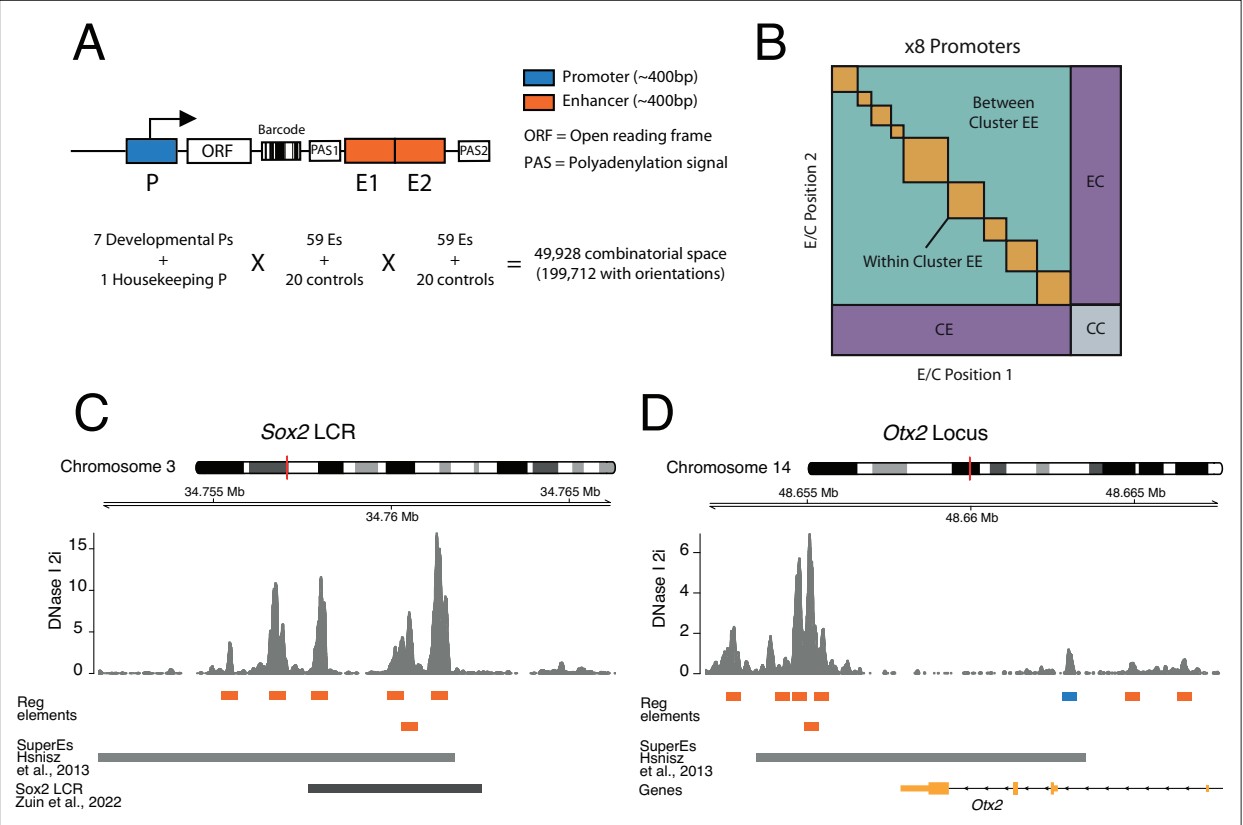

**Figure 1.** Schematic of three-way combinatorial approach. (**A**) Three-way combinatorial massively parallel reporter assay (MPRA) design to test enhancer-enhancer-promoter combinations. Eight barcoded reporter assay libraries, one per promoter, were constructed. Pairs of DNA elements (enhancers and scrambled control sequences) were inserted after barcoded reporter. The enhancers and controls can be placed in both orientations in either the enhancer position 1 (E1) or enhancer position 2 (E2). (**B**) The design of the library yields eight matrices that contain control-control combinations (CC), enhancer-control combinations (EC and CE), and enhancer-enhancer combinations (EE). (**C** and **D**) Two example loci, *Sox2* LCR (locus control region) (**C**) and *Otx2* (**D**) from where we selected enhancers to test in the reporter libraries. Enhancers (orange) were defined around DNAse I hypersensitivty sites from mouse embryonic stem cells (mESC). Promoters (blue) were chosen according to transcription start site (TSS) annotation and mESC DNAse I hypersensitivty sites. DNAse I data is from *Joshi et al., 2015*.

The online version of this article includes the following figure supplement(s) for figure 1:

**Figure supplement 1.** Schematic of cloning strategy to generate enhancer-enhancer-promoter libraries.

the transcription start site (TSS), because this includes the most information-rich part of functional promoters (*Martinez-Ara et al., 2022*; *van Arensbergen et al., 2017*).

## Clusters of enhancers selection

We selected clusters of enhancers that are either known to regulate the above promoters (*Blinka et al., 2016*; *Thomas et al., 2021*; *Zuin et al., 2022*; *Brosh et al., 2023*), or are candidates for such function because of their proximity to the respective promoters (see Methods and *Supplementary file 1*; *Hnisz et al., 2013*; *Whyte et al., 2013*). We also included putative enhancers from two additional developmental genes from the core pluripotency network (*Pou5f1 and Nodal*) (*Dunn et al., 2014*) that are located elsewhere in the genome.

## Definition and selection of single enhancers

We defined single enhancers as ~400 bp elements overlapping with DNase hypersensitive sites (DHSs) in clusters of enhancers (*Figure 1C and D*), as previously described (*Martinez-Ara et al., 2022*). Some of these enhancer clusters could be considered as single large enhancers (*Long et al., 2016*). However, since DHSs have been shown to retain enhancer activity individually (*Barakat et al., 2018*; *Agrawal et al., 2021*; *Bergman et al., 2022*; *Martinez-Ara et al., 2022*) we decided to test the

combinatorial contributions of these smaller elements of an enhancer cluster. In total we selected 59 of such single enhancers. We also included 20 random sequences as negative controls (see Methods, *Supplementary file 2*).

### Cloning of libraries

We cloned each promoter separately into the reporter assay vector. We then barcoded the eight separate vectors (see Methods). Next, we randomly combined enhancers and controls into pairs and cloned them into the eight different barcoded vectors downstream of the promoters (*Figure 1A* and *Figure 1—figure supplement 1*). This generated eight different EEP combinatorial libraries, one per promoter, with a total combinatorial space of 49,928 (8 promoters × 79 enhancers and controls × 79 enhancers and controls) or 199,712 if we consider the four possible orientation/position combinations of each enhancer or control fragment (*Figure 1A*). Of this space, the libraries covered 138,528 (69%) of the total combinatorial space, and after application of stringent quality filters (see Methods) we were able to determine activities of 110,180 (55%) combinations (*Supplementary file 3*).

### Explanation of library design

The design of the libraries generates eight matrices, one per promoter. These contain three types of combinations (*Figure 1B*): control-control (CC), enhancer-control (EC, regardless of position), and EE, which can be from the same or a different enhancer cluster. CC combinations are necessary to measure the baseline activity of each promoter. EC combinations are used to estimate the effects of single enhancers. In total, we measured 2584 CC combinations, 38,390 EC combinations, and 69,206 EE combinations.

### Library transfections

We transfected each library separately into mESCs, with three biological replicates performed on different days. RNA was extracted and barcodes were counted by sequencing as previously described (*Martinez-Ara et al., 2022*) both from RNA and from the plasmid libraries. We calculated the activity of each barcode as the ratio between barcode counts in cDNA and plasmid DNA. We then averaged barcode activities for each triple combination across a minimum of five barcodes. For these averaged activities, all developmental promoter libraries showed high correlations between replicates (Pearson's R=0.82–0.96) (*Figure 2—figure supplement 1*). The housekeeping promoter library showed somewhat lower correlations between replicates (Pearson's R=0.74–0.78), which is most likely due to a smaller dynamic range. For further analyses, we then averaged the biological replicates for each promoter.

## Effects of single enhancers

### Calculation of boost indices

In order to quantify the activation by each enhancer we first calculated the baseline activity of each promoter, defined as the median activity across all CC combinations. Then, for each individual enhancer-promoter combination, we calculated the median activity across all EC combinations. For each single enhancer we then calculated a boost index, which is the $\log_2$ ratio of its median activity across EC combinations over the promoter baseline across CC combinations (*Figure 2A*). We observed that there was little position and orientation dependence of the enhancer effects (*Figure 2—figure supplement 2*). We therefore averaged the boost indices of single enhancers across positions and orientations for further analyses.

### Selected DHSs are active enhancers

The single enhancer boost indices show that the majority of the sequences that we selected indeed act as enhancers, although the boost indices varied up to ~5 $\log_2$ units (*Figure 2A and B*). All tested enhancers showed significant activation of at least one promoter (at an estimated false discovery rate [FDR]<0.01) (*Figure 2B and D*). However, the dynamic ranges and effects of each enhancer vary per promoter, with the boost indices being particularly smaller for the housekeeping promoter (*Ap1m1*) (*Figure 2A and C*). We conclude that most DHSs that we selected from the enhancer clusters act individually as enhancers.

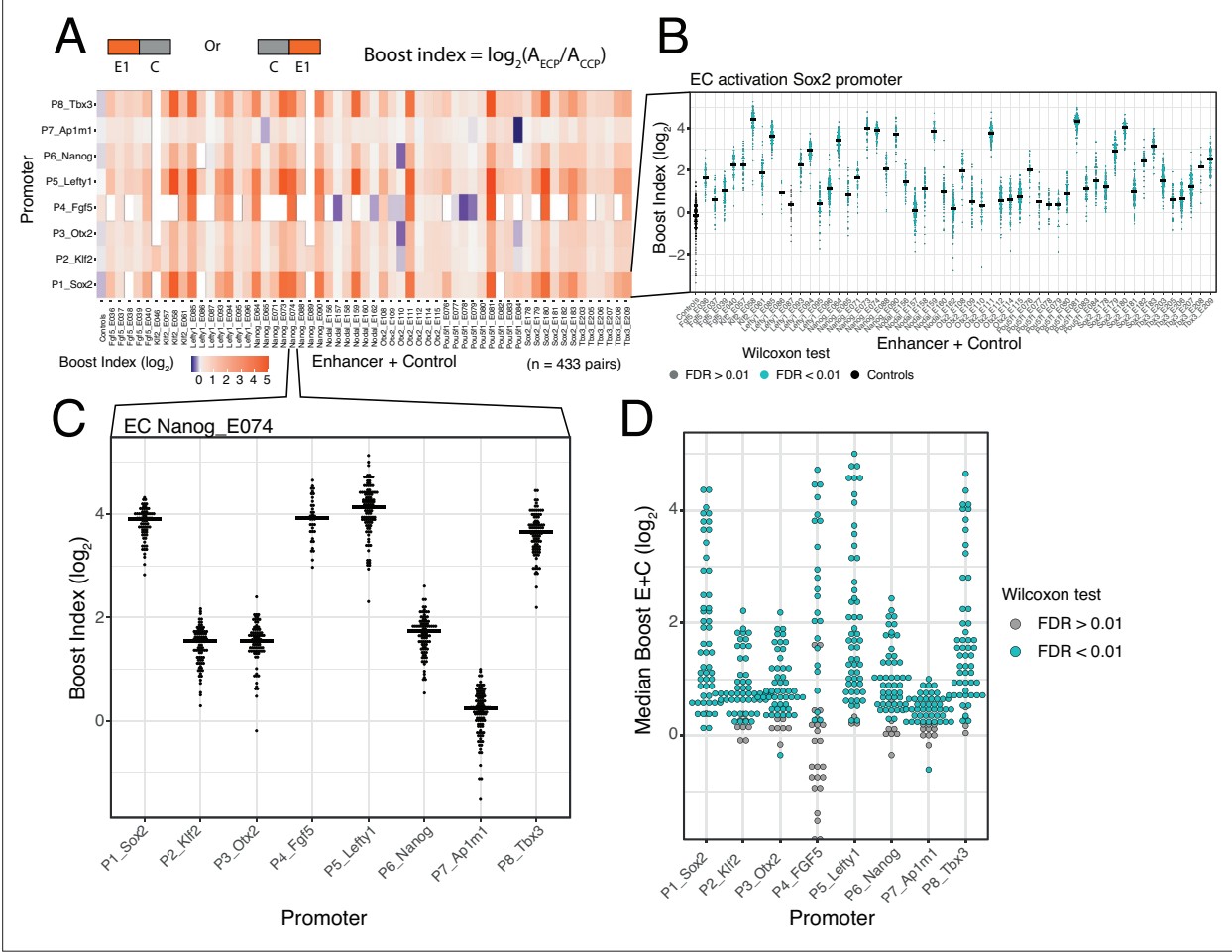

**Figure 2.** Effects of single enhancers across promoters. (**A**) Enhancer-promoter median boost index matrix of single enhancers. For each single enhancer a median boost index (see figure and Methods, $\log_2$(Activity_Combination/Activity_control-control_baseline)) was calculated across all enhancer-control combinations for that particular enhancer in any position and orientation, the baseline is the median activity of the promoter across all control-control combinations. For each control-control combination a boost index was calculated as the activity of the control-control combination over the median activity of all controls. A median control-control boost index was calculated based on this. Colour coding of the matrix corresponds to the median boost indexes, white spaces are missing data. (**B**) Distribution of boost indices for all enhancer+control combinations for the *Sox2* promoter. Leftmost column corresponds to the boost index distribution for all control-control combinations for the *Sox2* promoter. Each dot represents one enhancer+control combination. Horizontal lines correspond to the median of each distribution (same median as represented in the matrix in **A**). (**C**) Distribution of boost indices for all enhancer+control combinations for the enhancer Nanog_E074 across all promoters. Each dot represents one enhancer+control combination. Horizontal lines correspond to the median of each distribution. (**D**) Distribution of median boost indices for each single enhancer across promoters. Each dot represents the median of an enhancer across all enhancer+control combinations for that enhancers (same median as represented in the matrix in **A**). Colouring in B and D represents the significance at a 1% false discovery rate (FDR) for a Wilcoxon test comparing boost index distributions between enhancer+control combinations and the controls for each enhancer.

The online version of this article includes the following figure supplement(s) for figure 2:

**Figure supplement 1.** Reproducibility of experimental data matrix of replicate correlations for all eight enhancer-enhancer-promoter (EEP) libraries.

**Figure supplement 2.** Position and orientation bias.

**Figure supplement 3.** Selectivity of single enhancers.

## Promoter selectivity of enhancers

In our previous study of a large set of E-P pairs we found that the majority of tested enhancers exhibited significant preference for certain promoters (*Martinez-Ara et al., 2022*). Applying the same statistical analysis to the current enhancer set confirmed this finding: the variability of enhancer effects across promoters are generally larger than what is expected from experimental noise (*Figure 2—figure supplement 3*). This indicates that the single enhancers tested here show significant promoter

selectivity. This analysis also points to systematic differences in promoter responses; this will be further highlighted below.

## Enhancers generally combine near-additively

### EE combinations are generally stronger than single enhancers

We then focused on the effects of EE combinations. In the boost index matrix that covers all combinations for a single promoter, we observed that generally EE combinations were more active than EC combinations (*Figure 3A*). Indeed, quantitative analysis showed that EE combinations tend to induce stronger activation than the single enhancers (EC) (*Figure 3B and C*), although some saturation appears to occur with the strongest enhancers (*Figure 3C*).

### Additive model fits better than multiplicative model

We then asked how the effects of two single enhancers are integrated in the EE combinations. We considered two simple models: additivity and multiplicativity. For each EE combination we calculated the expected effect according to each of these models based on the single enhancer measurements, and then compared the observed and expected activities (additive in *Figure 3D and E*, multiplicative in *Figure 3F*, see Methods). Interestingly, for most of the promoters the additive model consistently showed a better fit to the measured data than the multiplicative model (*Figure 3E and F*, *Figure 3—figure supplement 1*). In particular, the multiplicative model tended to strongly overestimate activities at the higher range of activities (most active enhancers) for most promoters. When focusing on EE combinations for which the multiplicative and additive expected values differ more than 0.5 $\log_2$ units (to limit random noise effects), we found that the multiplicative model matched the observed values better for only 0–17% EE combinations across the seven promoters. Only for the *Ap1m1* promoter the multiplicative and additive models were nearly indistinguishable. This may be due to the low dynamic range of activities observed with this promoter. We note that for the *Fgf5* promoter in the low activity ranges the expected activities were consistently underestimated for both models. It is likely that this is an inaccuracy due to the relative sparseness of the *Fgf5* data (the baseline activity of this promoter was estimated from only 23 measured CC combinations, compared to 219–477 combinations for the other promoters). In summary, we conclude that a simple additive model fits the observed data better than a multiplicative model.

## Supra-additive activation is rare and promoter dependent

### Low frequency of supra-additive EE interactions

Most EE combinations show only minor deviations from the additive model (*Figure 3—figure supplement 1*), as illustrated by the distributions of observed/expected ratios (*Figure 4A*). By applying a simple error model to estimate the noise in the expected EE activities according to the additive model (*Figure 3D*, see Methods) we found that 74.5% of the measured EE activities lies within one standard deviation (SD) of the predicted activities (*Figure 4A*) and 94% lies within two SD. This underscores that most of the tested enhancers combine near-additively, and that both sub-additive and supra-additive effects are generally rare or weak.

### Supra-additivity occurs more frequently with weak promoters

Nevertheless, *Figure 4A* suggests that deviations from the additive model may vary between promoters. The *Sox2*, *Lefty1*, *Tbx3*, and *Fgf5* promoters showed a relatively large proportion of EE pairs with supra-additive effects, while the *Ap1m1* promoter exhibited the lowest frequency of supra-additivity. Interestingly, the proportion of supra-additivity correlates inversely with promoter baseline expression (*Figure 4B*, left panel). Thus, lowly active promoters allow more supra-additive EE interactions, although these remain a minority (<40% of EE pairs). In contrast, the proportion of sub-additive effects (i.e. >1 SD below expected activity) is very similar across promoters (*Figure 4B*, right). This latter observation suggests that there is no strong saturation effect that could explain the decrease in number of supra-additive interactions as promoter activity increases.

### EE supra-additive effects are relatively infrequent and often promoter-dependent

Next, we investigated whether supra-additivity involves a specific subset of enhancers. Visual inspection identified a few individual enhancers that exhibited supra-additive interactions with most other

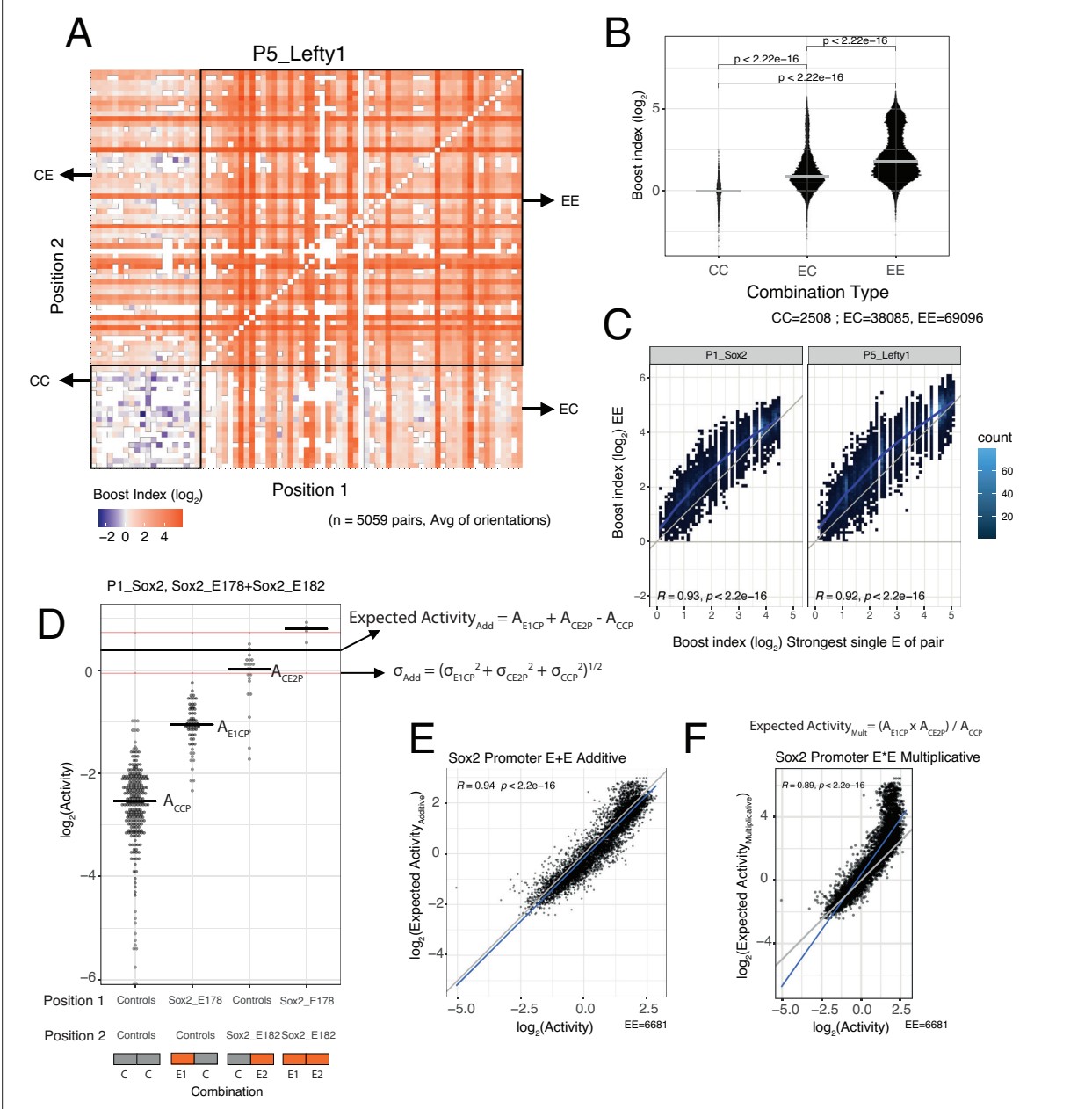

**Figure 3.** Effects of enhancer-enhancer (EE) combinations. (**A**) Fragment-fragment combinatorial boost index matrix for the *Lefty1* promoter. Each square represents one control-control (CC), enhancer-control (EC or CE), or EE combination. Colour coding corresponds to the average boost index for each combination across all orientations measured over the median control-control baseline. (**B**) Boost index distributions across all eight promoters for each combination type, control-control (CC), enhancer-control (EC regardless of position) or EE combinations. p-Values correspond to the result of a Wilcoxon test. (**C**) Relationship between observed boost index for each EE combination and the observed boost index of the strongest single enhancer of the pair for *Sox2* and *Lefty1* promoters. Blue lines represent the LOESS fit of the data. (**D**) Observed and expected additive activities for the Sox2_E178+Sox2_E182 combination with the *Sox2* promoter and the individual activities of each of the elements. Each column represents the observed activities for the control-contol combinations, the enhancer-control combinations, and the EE combinations. The horizontal bars represent the median of each distribution. The horizontal black line represents the expected additive activity of the EE combination as calculated by the formula in the panel in the linear space. The horizontal red lines represent the propagated standard deviations of the expected additive activity of the EE combinations as calculated by the formula in the panel. (**E and F**) Relationship between observed and expected activities (additive in **E**, multiplicative in **F**) for all EE combinations for the *Sox2* promoter. The blue lines represent the linear fit of the data. Grey diagonal line is the x=y identity line. In all panels R represents Pearson's correlation coefficient. Expected activities are calculated in the linear space and then plotted in the log₂ space.

The online version of this article includes the following figure supplement(s) for figure 3:

**Figure supplement 1.** Additivity versus multiplicativity for all promoters.

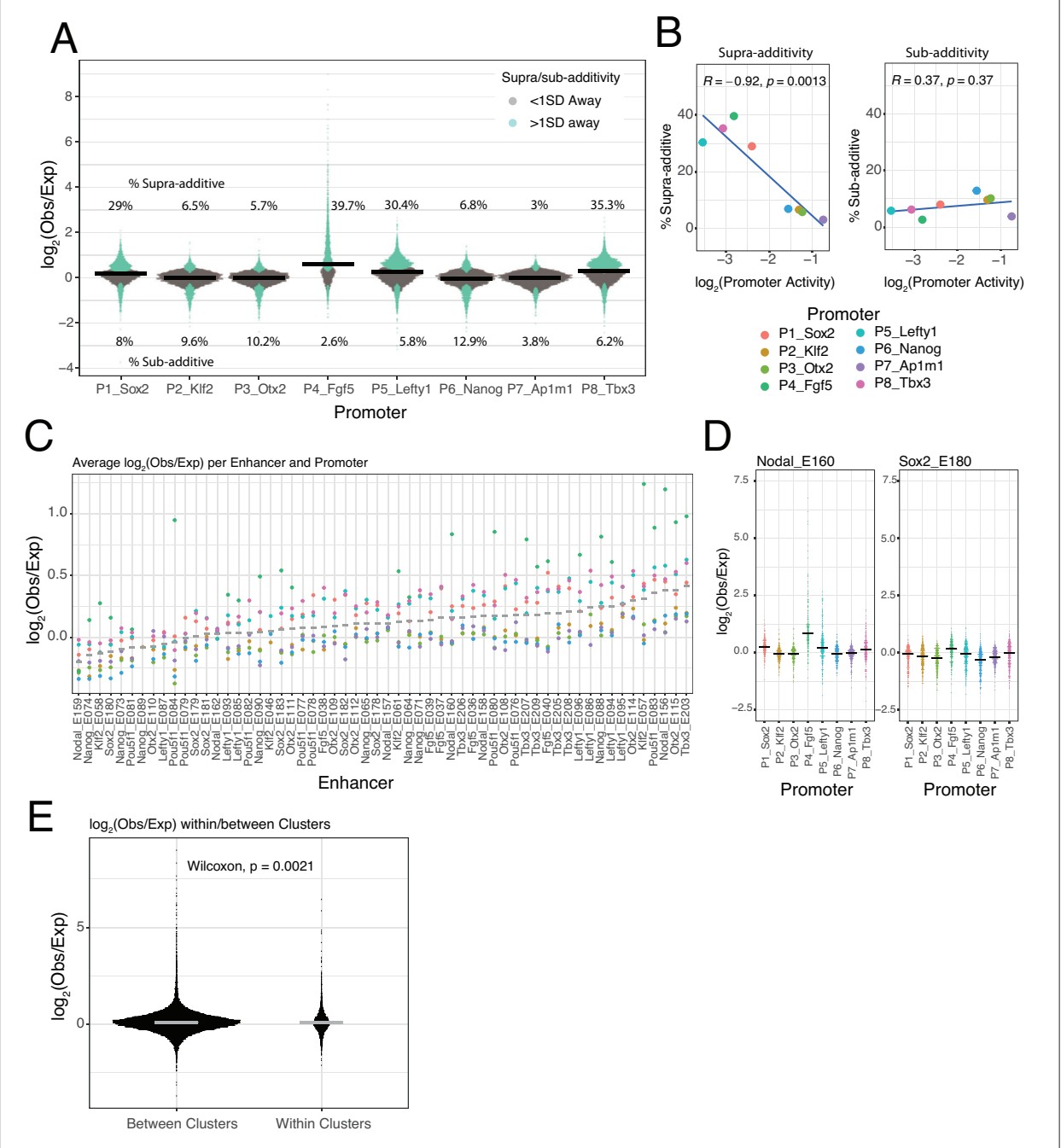

**Figure 4.** Supra- and sub-additive behaviours of enhancer combinations. (**A**) Distributions of log$_2$ observed activities over expected additive activities ratios of enhancer-enhancer combinations across promoters. Coloured in turquoise are supra- and sub-additive combinations for which the observed activity is more than one standard deviation away from the expected activity. Horizontal bars represent the median of each distribution. Numbers on the top part are the percentage of supra-additive combinations for each promoter. Numbers on the lower part are the percentage of sub-additive combinations for each promoter. (**B**) Relationship between the percentages of supra- and sub-additive enhancer-enhancer combinations and promoter control-control baselines. Blue lines are the linear fit of the data. R is Pearson's correlation. (**C**) Average supra- or sub-additive behaviour of each single enhancer across enhancer-enhancer combinations for each promoter. Each dot represents the median log$_2$ observed over expected for all enhancer-enhancer combinations of a single enhancer and a particular promoter. Grey bars represent the median of each distribution. (**D**) For two example enhancers, distribution of log$_2$ observed over expected ratios for combinations of that enhancer with any other enhancer and promoter. Horizontal black bars represent the median of the distribution. (**E**) Distribution of log$_2$ observed over expected ratios for enhancer-enhancer pairs from the same enhancer cluster (within clusters) or from different enhancer clusters (between clusters).The p-value results from comparing both distributions using a Wilcoxon test. Horizontal grey bars represent the median of each distribution. In all panels Obs/exp refers to observed activity over expected additive activity.

enhancers, but often in the context of particular promoters. An example is Nodal_E160 in combination with the *Fgf5* promoter (*Figure 4D*, left panel). Other enhancers showed a diversity of supra-additive activities. Out of the 58 tested enhancers 32.7% (19/58) showed consistent supra-additivity in at least one promoter context (>0.5 log2(observed/expected), mean over all enhancers in at least one promoter context), and only 10.3% (6/58) showed consistent supra-additivity in two or more promoter contexts (*Figure 4C*). However, most enhancers behave like Sox2_E180 (*Figure 4D*, right panel), showing small deviations from the additive behaviour with most other enhancers, and with all of the tested promoters (*Figure 4C*).

## No evidence for locus-specific EE supra-additivity

We then asked whether enhancer pairs derived from the same genomic locus have a higher propensity towards supra-additivity than those not derived from the same locus. This could point towards co-evolution of enhancers that jointly control a gene. However, we found that EE combinations from the same locus showed on average not a higher observed/expected ratio (according to the additive model) than EE combinations from different loci; in fact, this ratio is even slightly lower (*Figure 4E*). Thus, generally there appears to be no preferential synergy among enhancers that are located in the same genomic locus.

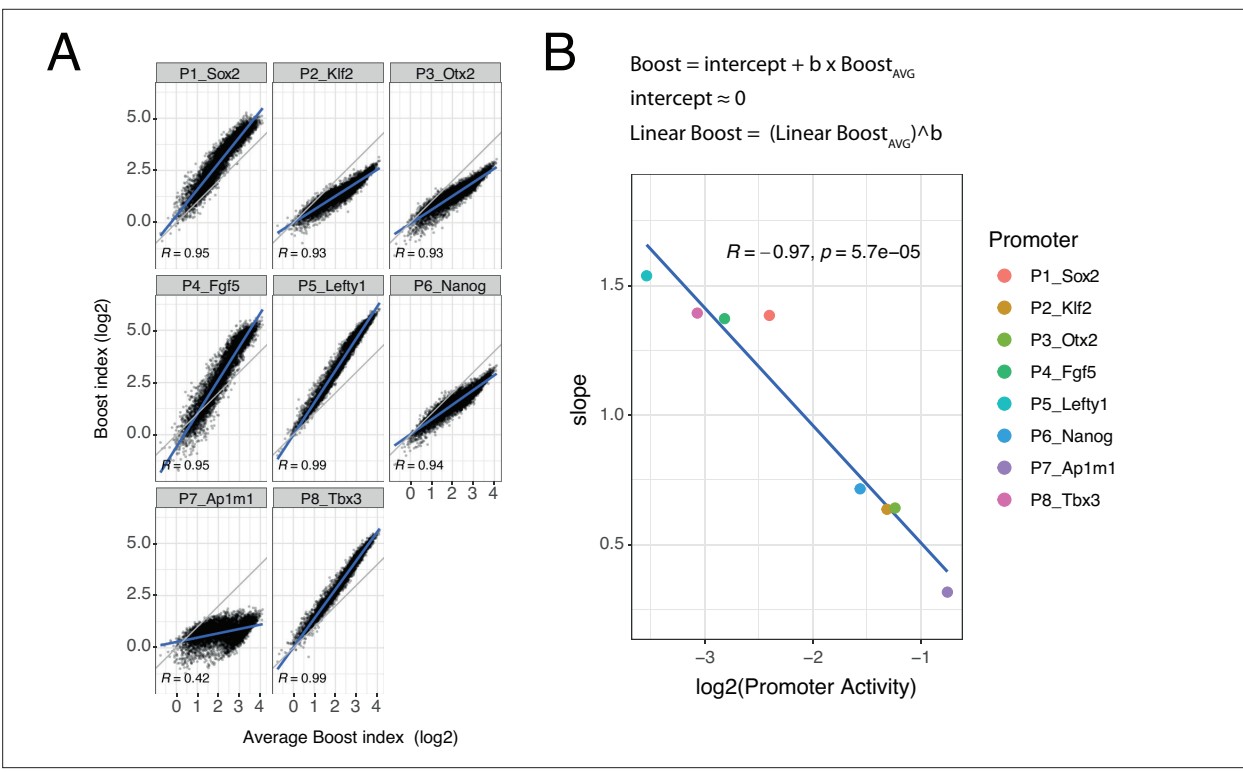

**Figure 5.** Non-linear responses of promoters to enhancer-enhancer combinations. (**A**) Relationship between observed boost indices and average boost index across promoters for all shared enhancer-enhancer combinations. Blue lines represent the linear fit of the data. (**B**) Relationship between the slopes extracted from the linear fits in **A** and the baseline promoter activities derived from the control-control combinations. The formulae depict the relationship between the average boost indices and the observed boost indices of each promoter through the extracted slopes. For both panels R is Pearson's correlation coefficient.

The online version of this article includes the following figure supplement(s) for figure 5:

**Figure supplement 1.** Promoter-promoter boost index correlations for all shared enhancer-enhancer combinations.

**Figure supplement 2.** Non-linear responses of promoters to enhancer-control (single enhancers) combinations.

## Enhancer effects scale non-linearly across developmental promoters

### Enhancer effects correlate well across developmental promoters

Prompted by our observations that some supra-additive EE interactions appear to be more pronounced in the context of specific promoters, we decided to systematically compare the boost indices of all EE pairs between the eight promoters. Strikingly, all seven developmental promoters showed strong pairwise correlations of their boost indices (Pearson's R=0.79–0.98), whereas the housekeeping promoter (*Ap1m1*) correlated much less with each of the developmental promoters (Pearson's R=0.19–0.60; *Figure 5—figure supplement 1*).

### Slopes of promoter boost responses differ

Although the developmental promoters showed strong correlations, we were surprised to notice that the slopes of these relationships often deviated from 1. To further explore this, we plotted the boost indices of each promoter against the average boost indices across all eight promoters (*Figure 5A*). All of the developmental promoters correlated strongly with this average pattern (Pearson's R=0.93–0.99) but this was not the case for the housekeeping promoter (Pearson's R=0.42). The slopes relative to the averaged promoters ranged from 1.54 to 0.32, i.e., about fivefold, across the developmental promoters. These results imply that the developmental promoters differ in the scaling (i.e. steepness) of their responses to the set of EE pairs, even though the pattern of their responses is highly similar. Interestingly, the relative slopes negatively correlate with the baseline activity of these promoters (*Figure 5B*). Thus, promoters with low intrinsic activity respond more steeply to changes in combined EE activity than promoters with high intrinsic activity.

### Promoter response differences also occur with single enhancers

Finally, we asked whether these differential responses of promoters require EE pairs, or can also be observed with single E elements. Indeed, this is the case (*Figure 5—figure supplement 2*).

## Discussion

### A large-scale survey of EEP combinations

Here, we employed an MPRA approach to measure three-way regulatory interactions in ~69,000 EEP combinations involving 8 promoters and 59 enhancers. This is of a much larger scale than previous efforts to elucidate the combinatorial interplay among enhancers, which typically focused on a handful of enhancers in single genomic loci (*Bothma et al., 2015*; *Hay et al., 2016*; *Shin et al., 2016*; *Dukler et al., 2016*; *Joo et al., 2016*; *Lam et al., 2015*; *Carleton et al., 2017*; *Moorthy et al., 2017*; *Osterwalder et al., 2018*; *Thomas et al., 2021*; *Lin et al., 2022*; *Brosh et al., 2023*). These previous studies pointed to a diversity of functional EE interactions, including redundancy, additivity, and synergy. Our study also uncovered a broad spectrum of functional interactions, but simple near-additive effects of EE combinations were predominant, while supra-additive interactions occurred at a lower frequency and were dependent on the specific enhancer and promoter context.

### Promoters 'interpret' and scale EE interplay

An important finding is that promoters can have substantial impact on supra-additivity of EE interactions. Promoters may thus be seen as 'interpreters' of EE combinations. This underscores the importance of studying triple combinations of elements, and perhaps in the future even more complex settings. A salient finding is that the different promoters show different degrees of responsiveness to EE pairs, with the stronger promoters being boosted less than the weaker promoters. The responses of the promoters follow roughly a power-law function of which the exponent is dependent on the promoter (*Figure 5A* and *Figure 5—figure supplement 2*, which are on log-log scale). Because it is not possible to define an absolute measure of enhancer strength, we cannot determine the absolute values of the exponents of the promoter power-law responses. This would be of interest because exponents <1 would lead to 'diminishing returns' with higher EE activities, while exponents >1 would correspond to increasing non-linear (see definition of non-linear in Computational methods section) amplification with higher EE activities. The apparent saturation observable at high enhancer activities in *Figure 3C* suggests that the exponents are most likely below 1. Our findings resemble those of

an earlier study (*Hong and Cohen, 2022*), indicating that promoter activities are non-linearly scaled by the genomic environment, with the steepness of the scaling being negatively correlated with promoter strength.

## Episomal MPRAs versus genomic context

The reporters in our MPRA were not integrated into the genome. On the one hand, this ensures that all EEP combinations are tested under similar conditions, without confounding effects of variable chromatin contexts. On the other hand, we cannot rule out that the results are not fully representative of a natural chromatin environment. However, other studies have indicated that transiently transfected reporters generally behave quite similar, although not identically, to integrated reporters (*Inoue and Ahituv, 2015*; *Klein et al., 2020*), and another study indicated that chromatin context can scale transcriptional activity of inserted reporters in a simple linear manner that is largely independent of the inserted sequence (*Maricque et al., 2018*).

## Results may be biased by model system

We cannot rule out that our results are biased by our selection of enhancers and promoters, which was mostly focused on genes linked to pluripotency in mouse ESCs. We may have missed more prominent synergistic or other non-additive effects that may arise in different cell types or among different classes of enhancers and promoters. However, the protocol that we developed for three-way MPRAs should be generally applicable to any set of enhancers and promoters.

## Spacing and context of elements

It should also be considered that in our MPRA the enhancer and promoter elements are placed in fixed positions and close to one another, which does not recapitulate their relative positioning in the genome. Genomic distance and chromatin contexts could act as thresholds for activation, or turn additive interactions into non-linear relationships. For instance, the ability of a *Sox2* enhancer to activate its cognate promoter was found to depend on their distance, following a non-linear decay function (*Zuin et al., 2022*). The rules for EE interplay may depend on the EE distance or genomic context, as suggested by a recent report indicating that distant pairs of enhancers are more often synergistic than closely spaced enhancers, which tend to act additively (*Lin et al., 2022*). A recent study reported synergies between E elements in the *Sox2* locus control region (a large compound enhancer region) in the native genomic context (*Brosh et al., 2023*). We did not observe such synergies, but the data of that study are not directly comparable to ours due to differences in the size of the elements tested. In the native *Fgf5* locus, several of the E elements exhibited some degree of synergy in differentiated cells (*Thomas et al., 2021*). However, in mESCs this synergy was not detected, which is in line our results for the same E elements from the *Fgf5* locus. Possibly, the default mode of EE interplay is additive, while the precise chromatin context, element spacing, and cellular state may enable synergy in some genomic loci.

## Enhancer-promoter compatibility re-interpreted

In our previous MPRA study (*Martinez-Ara et al., 2022*) we found evidence for selectivity of enhancer-promoter functional interactions (see definition of selectivity in Computational methods section). However, due to the limited number of enhancer-promoter combinations tested we could not fully disentangle what caused these differences in promoter responses. The data from the eight promoters indicate that there are two intermingled factors that account for differences between promoters. First, distinct classes of promoters show different specificities, as illustrated by the poor correlations between the *Ap1m1* housekeeping promoter and the developmental promoters. Second, promoters scale the input they receive from enhancer pairs differently, as discussed above. It is probable that the compatibility differences we observed in our previous study (*Martinez-Ara et al., 2022*) are a mix of these two phenomena: (1) promoters that respond differently to enhancers because of specificity, and (2) promoters that respond differently to enhancers because of differences in responsiveness caused by their intrinsic strength.

## Relevance of findings and context

The findings presented here offer insights into two aspects of the gene regulatory process – interplay between enhancers, and promoter integration of the resulting signals – in a more quantitative manner.

The near-additivity of enhancers will help us understand how different enhancers may be more or less relevant in different loci. The non-linearity of promoter responsiveness and the effects of promoters on enhancer-enhancer interactions illustrates how some promoters may be more sensitive to activating signals. Thus, it will be useful to quantify promoter responsiveness genome-wide to understand which genes may be more or less sensitive than others to environmental cues or to non-coding mutations. These aspects may be especially relevant for developmentally regulated transcription factors and their target genes, like the ones here studied, as they can be very sensitive to dosage changes (*Naqvi et al., 2023*).

# Methods

## Key resources table

| Reagent type (species) or resource | Designation | Source or reference | Identifiers | Additional information |
|---|---|---|---|---|
| Cell line (*Mus musculus*) | E14tg2a mouse embryonic stem cell (mESC) | ATCC | CRL-1821 | |
| Gene (*M. musculus*) | Klf2 | GenBank | 16598 | Promoters and enhancers; coordinates in *Supplementary file 1* |
| Gene (*M. musculus*) | Sox2 | GenBank | 20674 | Promoters and enhancers; coordinates in *Supplementary file 1* |
| Gene (*M. musculus*) | Otx2 | GenBank | 18424 | Promoters and enhancers; coordinates in *Supplementary file 1* |
| Gene (*M. musculus*) | Lefty1 | GenBank | 13590 | Promoters and enhancers; coordinates in *Supplementary file 1* |
| Gene (*M. musculus*) | Fgf5 | GenBank | 14176 | Promoters and enhancers; coordinates in *Supplementary file 1* |
| Gene (*M. musculus*) | Tbx3 | GenBank | 21386 | Promoters and enhancers; coordinates in *Supplementary file 1* |
| Gene (*M. musculus*) | Nanog | GenBank | 71950 | Promoters and enhancers |
| Gene (*M. musculus*) | Ap1m1 | GenBank | 11767 | Promoters and enhancers |
| Recombinant DNA reagent | Downstream Assay vector (JvAp102) | *Martinez-Ara et al., 2022* | JvAp102 | Downstream reporter assay plasmid; 'see Methods |
| Sequence-based reagent | Enhancer and promoter oligos | This paper | Primers | *Supplementary file 4* |
| Sequence-based reagent | Barcoding primers and sequencing primers | *Martinez-Ara et al., 2022* | Barcoding reagents | *Supplementary file 4* |
| Sequence-based reagent | Synthetic DNA negative controls | This paper | Negative controls | *Supplementary file 2* |
| Peptide, recombinant protein | Gibson Assembly Master Mix | NEB | Cat# E2611S | |
| Peptide, recombinant protein | I-CeuI | NEB | #R0699S | |
| Peptide, recombinant protein | I-SceI | NEB | #R0694S | |
| Commercial assay or kit | Mouse Embryonic Stem Cell Nucleofector Kit | Lonza | #VPH-1001 | |
| Chemical compound, drug | TRIsure | Bioline | (#BIO-38032) | |
| Commercial assay or kit | GeneJET RNA extraction kit | Thermo Fisher | #K0732 | |
| Peptide, recombinant protein | DNase I | Roche | #04716728001 | |
| Peptide, recombinant protein | Maxima Reverse transcriptase | Thermo Fisher | #EP0743 | |
| Peptide, recombinant protein | MyTaq Red mix | Bioline | #BIO-25043 | |
| Strain, strain background (*Escherichia coli*) | e. cloni 10G supreme | Lucigen | #60081-1 | Electrocompetent cells |
| Peptide, recombinant protein | Takara ligation kit version 2.1 | Takara | 6022 | |
| Peptide, recombinant protein | Klenow HC 3'->5' exo | NEB | #M0212L | |
| Commercial assay or kit | ISOLATE II PCR and Gel Kit | Bioline | BIO-52059 | |
| Peptide, recombinant protein | Fast-link ligase | Lucigen | LK0750H | |
| Peptide, recombinant protein | End-It DNA End-Repair Kit | Epicentre | #ER0720 | |
| Commercial assay or kit | dsDNA High sensitivity Qubit kit | Invitrogen | #Q33231 | |

*Continued on next page*

*Continued*

| Reagent type (species) or resource | Designation | Source or reference | Identifiers | Additional information |
|---|---|---|---|---|
| Commercial assay or kit | CleanPCR magnetic beads | CleanNA | #CPCR-0050 | |
| Peptide, recombinant protein | XcmI | NEB | #R0533S | |
| Peptide, recombinant protein | T4 DNA ligase | Roche | #10799009001 | |
| Peptide, recombinant protein | AvrII | Thermo Fisher | #ER1561 | |
| Peptide, recombinant protein | NheI | NEB | #R0131S | |
| Strain, strain background (*Escherichia coli*) | 5-alpha Competent *E. coli* | NEB | #C2987 | Competent cells |
| Peptide, recombinant protein | LIF | Sigma-Aldrich | #ESG1107 | 2i+LIF media |
| Chemical compound, drug | Monothioglycerol | Sigma-Aldrich | #M6145-25ML | 2i+LIF media |
| Chemical compound, drug | CHIR-99021 | MedChemExpress | #HY-10182 | 2i+LIF media |
| Chemical compound, drug | PD0325901 | MedChemExpress | #HY-10254 | 2i+LIF media |
| Other | BSA | Gibco | #15260-037 | 2i+LIF media |
| Other | DMEM-F12medium | Gibco | #11320-033 | 2i+LIF media |
| Other | Neurobasal medium | Gibco | #21103-049 | 2i+LIF media |
| Chemical compound, drug | N27 | Gibco | #17504-044 | 2i+LIF media |
| Chemical compound, drug | B2 | Gibco | #17502-048 | 2i+LIF media |
| Commercial assay or kit | MycoAlert Mycoplasma Detection Kit | Lonza | #LT07-318 | |
| Software, algorithm | BatchPrimer3 version 1.0 | *You et al., 2008* | version 1.0 | https://wheat.pw.usda.gov/demos/BatchPrimer3/ |
| Software, algorithm | Starcode | *Zorita et al., 2015* | version 1.1 | https://github.com/gui11aume/starcode |
| Software, algorithm | Python | *Rossum and Drake, 2009* | version 3.6 | https://www.python.org/downloads/release/python-362/ |
| Software, algorithm | Bowtie2 | *Langmead and Salzberg, 2012* | version 2.3.4 | http://bowtie-bio.sourceforge.net/bowtie2/index.shtml |
| Software, algorithm | R | *R Development Core Team, 2021* | version 4.0.5 | https://www.r-project.org/ |
| Software, algorithm | ggplot2 | *Wickham, 2016* | ggplot2 | https://ggplot2.tidyverse.org/ |
| Software, algorithm | Snakemake | *Köster and Rahmann, 2012* | version 4.4.0 | https://anaconda.org/bioconda/snakemake/files?version=4.4.0 |

## Cell culture

We used E14tg2a male mESCs (ATCC CRL-1821) cultured in 2i+LIF for all experiments. As described (*Martinez-Ara et al., 2022*, 35594855), we prepared 2i+LIF media according to the 4DN nucleome protocol (https://data.4dnucleome.org/protocols/cb03c0c6-4ba6-4bbe-9210-c430ee4fdb2c/). The reagents used were Neurobasal medium (#21103-049, Gibco), DMEM-F12medium (#11320-033, Gibco), BSA (#15260-037; Gibco), N27 (#17504-044; Gibco), B2 (#17502-048; Gibco), LIF (#ESG1107; Sigma-Aldrich), CHIR-99021 (#HY-10182; MedChemExpress) and PD0325901 (#HY-10254; MedChem-Express), monothioglycerol (#M6145-25ML; Sigma), and glutamine (#25030-081, Gibco). Mycoplasma contamination was ruled out by monthly tests (#LT07-318; Lonza).

## Selection of promoters, enhancers, and controls

For the construction of the libraries we selected seven developmental genes from the mESC pluripotency regulatory network (*Acampora et al., 2013*; *Dunn et al., 2014*; *Kim et al., 2014*; *Thomas et al., 2021*), *Klf2, Sox2, Nanog, Otx2, Lefty1, Ffg5, Tbx3*; and the housekeeping gene *Ap1m1* (*Hounkpe et al., 2021*), which is neighbouring the *Klf2* gene. The promoters of these genes were identified based on transcript and TSS annotation (*Frankish et al., 2019*) and overlaps with DHS (*Joshi et al., 2015*) as previously described (*Martinez-Ara et al., 2022*). We then selected sequences that approximately cover –350 to 50 bp around the TSS.

The clusters of enhancers were selected in as follows: (1) Clusters of enhancers that were known to regulate the developmental genes (*Nanog, Sox2, Fgf5* clusters) (*Blinka et al., 2016*; *Brosh et al.,*

*2023*; *Thomas et al., 2021*; *Zuin et al., 2022*). (2) Putative clusters of enhancers in proximity of one of the developmental genes and that overlapped with previously defined super-enhancers (*Lefty1, Tbx3, Otx2* clusters) (*Hnisz et al., 2013*; *Whyte et al., 2013*). (3) A cluster of putative enhancers previously identified by us (*Martinez-Ara et al., 2022*) to act as enhancers in an MPRA (*Klf2* cluster). (4) Clusters of enhancers identified as putative enhancers of other mESC pluripotency genes for which we did not include their promoters (*Pou5f1* and *Nodal* clusters) and overlapping with previously defined super-enhancers (*Hnisz et al., 2013*; *Hnisz et al., 2015*; *Whyte et al., 2013*).

Clusters of enhancers were divided into single enhancers based on DHS (*Joshi et al., 2015*). These single enhancers were then resized to ~450 bp from the centre of DNAse hypersensitivity peaks as previously described (*Martinez-Ara et al., 2022*).

For the library construction we included twenty 450-bp-long randomised control sequences as previously described (*Martinez-Ara et al., 2022*). Fifteen of these were based on five DHS peak sequences that were randomly scrambled. The other five controls were randomly generated sequences with the similar GC content as the DHS peak sequences. These control sequences were ordered as synthetic DNA. *Supplementary file 2* contains the sequences of these controls.

For single enhancers and promoters we designed PCR primers against the 50 bp ends of each selected element using BatchPrimer3 version 1.0 (*You et al., 2008*). This yielded PCR products of ~400 bp for each element. *Supplementary file 1* lists the coordinates of all individual enhancers and promoters. *Supplementary file 4* lists primer sequences.

## EEP library generation

The vector for library construction was previously used for the downstream reporter assay in our previous study (*Martinez-Ara et al., 2022*). This vector is based on a pSMART backbone. It contains a green fluorescent protein (GFP) open reading frame followed by a barcode, and a psiCheck polyadenylation signal (PAS) introduced during barcoding, followed by the cloning site for inserts and a triple polyadenylation site (SV40+bGH+psiCheckPAS).

Each of the selected eight promoters were amplified by PCR and individually inserted by Gibson assembly (#E2611S; New England Biolabs) into the reporter vector. Each construct was transformed into standard 5-alpha competent bacteria (#C2987; NEB) grown overnight in 500 ml of standard Luria Broth (LB) with 50 mg/ml of kanamycin and purified. Then, each promoter sequence was verified by Sanger sequencing.

Each of the promoter vectors was then barcoded. Similar to what we described previously (*Martinez-Ara et al., 2022*)**,** we digested 10 µg of each vector with AvrII (#ER1561; Thermo Fisher) and XcmI (#R0533; NEB) and performed a gel-purification. Barcodes were amplified in 10 separate 100 µl PCRs using 5 µl of 10 mM primer 275JvA, 5 ml of 10 mM primer 465JvA, and 1 µl of 0.1 mM template 274JvA (see *Supplementary file 4* for sequences). 14 PCR cycles were performed using MyTaq Red Mix (#BIO-25043; Bioline), yielding 30 µg of barcodes. We purified barcodes by phenol-chloroform extraction and isopropanol precipitation. Barcodes were digested overnight with 80 units of NheI (#R0131S; NEB) and purified by magnetic bead purification (#CPCR-0050; CleanNA). Each promoter vector and barcodes were then ligated in one 100 µl reaction. In each reaction we used 3 µg of digested vector and 2.7 µg digested barcodes, 20 units NheI (#R0131S; NEB), 20 units AvrII, 10 ml of 103 CutSmart buffer, 10 µl of 10 mM ATP, 10 units T4 DNA ligase (#10799009001 Roche). We performed a cycle-ligation of six cycles (10 min at 22°C and 10 min at 37°C), followed by 20 min heat-inactivation at 80°C. Linear barcoded vectors were purified by magnetic beads and digested for 3 hr with 40 units of XcmI (#R0533S; NEB). Finally barcoded vectors were purified again by gel-purification.

Single enhancers were amplified by PCR from E14tg2a mESC genomic DNA, then purified with magnetic beads (#CPCR-0050, Clean NA) and quantified by Qubit using dsDNA High sensitivity Qubit kit (#Q33231; Invitrogen). Single enhancers and random controls were combined in approximately equimolar manner. This pool of elements was end-repaired using End-It DNA End-Repair Kit (#ER0720; Epicentre) and self-ligated using Fast-link ligase (LK0750H; Lucigen). Duplets of 800–1000 bp were excised from agarose gel, purified (BIO-52059; Bioline), and A-tailed using Klenow HC 3′ → 5′ exo (#M0212L; NEB).

The pool of duplets was ligated to each of the barcoded promoter vectors using Takara ligation kit version 2.1 (#6022; Takara). Ligation products were purified using magnetic bead purification and 2 µl of the ligation were electroporated into 20 µl of electrocompetent e. cloni 10G supreme (#60081-1;

Lucigen). To estimate the approximate complexity of the resulting libraries, dilutions of the electroporated bacteria were plated for overnight culture and colonies were counted. We aimed at a minimum of 2 million complexity per library in order to have sufficient expected representation of all possible combinations.

When the estimated complexity of the libraries was not high enough, we amplified the ligation products with primers aligning to the vector. This generated PCR products consisting of the duplets pool and a vector overhang. These PCR products were used for Gibson assembly with the barcoded promoters (#E2611S; NEB). Gibson assembly reactions were purified with magnetic beads and 2 µl of the reaction were electroporated into 20 µl of electrocompetent e. cloni 10G supreme (#60081-1; Lucigen).

Each library was grown overnight in 500 ml of standard LB with 50 mg/ml of kanamycin and purified using a maxiprep kit (K210016, Invitrogen).

## Characterisation of libraries by iPCR and sequencing

Barcode to insert (enhancer and control duplets) combinations were identified by inverse PCR (iPCR) and Illumina sequencing as described before (*van Arensbergen et al., 2017*). In brief, each library was digested overnight with I-CeuI (#R0699S, NEB), barcode-insert fragments were then circularised, remaining linear fragments digested, and barcode-insert fragments were linearised again with I-SceI (#R0694S; NEB). Barcode-insert fragments were then amplified by PCR with Illumina adapters and sequenced on an Illumina NextSeq 550 platform using 150 bp paired-end sequencing. Each library was processed separately and mixed together for sequencing.

## Transfection of libraries

Each EEP library was transfected separately into E14tg2a mESCs. Per library 20 million cells were nucleofected (5 µg of library and 5 million cells per cuvette, 5 cuvettes) using Amaxa nucleofector II, program A-30, and Mouse Embryonic Stem Cell Nucleofector Kit (#VPH-1001, Lonza). Three biological replicates were performed per library on different days. Cells were collected 24 hr after transfection in 5 ml of TRIsure (#BIO-38032; Bioline) and frozen at –80°C until further processing.

## cDNA sequencing

RNA was extracted and prepared for sequencing as described for the Downstream Assay in our previous study (*Martinez-Ara et al., 2022*). From each sample, the aqueous phase containing the total RNA of the TRIsure solution was extracted and purified on an RNA extraction column (#K0732, Thermo Scientific). Total RNA was digested with 10 units of DNase I (#04716728001; Roche) for 30 min in 10-8 10 µl reactions containing 5 µg of RNA. DNase I was inactivated by the addition of 1 µl of 25 mM EDTA and incubation at 70°C for 10 min.

cDNA was produced in 20 µl reactions (1 per 10 µDNAse reaction) using Maxima Reverse transcriptase (#EP0743; Thermo Fisher Scientific) and a gene-specific (targeting the GFP reporter) primer (JvA304, see *Supplementary file 4* for sequence). dNTPs and the primer were mixed with the DNAse digested RNA and incubated at 65°C for 5 min. The RT buffer, enzyme, and RNAse inhibitor were added and the reaction was incubated at 50°C for 1 hr. cDNA was then amplified by two nested PCRs to make it strand specific. The first PCR uses index variants of 285JvA (containing the S2, index, and p7 adaptor) and primer 305JvA (targeting the adapter introduced by 304JvA). Each 20 µl RT reaction was amplified in a 100 µl PCR with MyTaq Red mix (#BIO-25043; Bioline). The second PCR was performed using 10 µl of the product of the previous reaction in a 100 µl reaction using the same index variant primer and index variants of 437JvA (containing the S1, index, and p5 adaptor) (see *Supplementary file 4* for primer sequences). The first PCR was run for 10 cycles and the second one for 8 cycles using the recommended Mytaq Red mix conditions (for both PCRs: 1 min 96°C, then each cycle 15 s 96°C, 15 s 60°C, 15 s 72°C). PCR products were purified with magnetic beads and mixed for sequencing in an Illumina NovaSeq 6000 platform using 100 bp single-end reads.

## pDNA sequencing

Plasmid libraries were processed for sequencing as described (*Martinez-Ara et al., 2022*) but using dual indexing for sequencing on a NovaSeq 6000 platform with 100 bp single-end reads. 1 µg of each plasmid was digested with I-SceI in order to linearise the plasmid. Barcodes were amplified by PCR for

9 cycles from 50 ng of material. Primers and reaction conditions were the same as in the amplification of cDNA. PCR products were purified using magnetic beads and mixed for sequencing.

## Computational methods

cDNA and plasmid DNA (pDNA) data pre-processing and linking of barcodes to duplets was performed using a custom snakemake pipeline (*Köster and Rahmann, 2012*). All other data analyses and quantifications were performed in R (*R Development Core Team, 2021*). Figures were generated using ggplot2 (*Wickham, 2016*). The snakemake pipeline and all the scripts used in this publication are available in a GitHub repository (https://github.com/vansteensellab/EEPCombinations, copy archived at *Martinez-Ara, 2024*).

## Linking barcodes to duplet inserts

The custom pipeline for linking barcodes to duplet inserts was previously described (*Martinez-Ara et al., 2022*, 35594855). In brief, iPCR reads from each library were locally aligned using Bowtie (version 2.3.4) (*Langmead and Salzberg, 2012*) with very sensitive parameters (–very-sensitive-local) on a custom Bowtie genome. This custom Bowtie genome consists of virtual chromosomes that correspond to each of the enhancer and control sequences used to generate the combinatorial libraries. Bam files were processed by a custom Python script (*Rossum and Drake, 2009*). This script extracts from read 1 the barcode sequence, and the identity and orientation of the DNA fragment in position 1. From read 2 it extracts the identity and orientation of the DNA fragment in position 2. Finally, barcodes were clustered using Starcode (version 1.1) (*Zorita et al., 2015*) to remove PCR and sequencing errors.

## cDNA and pDNA data pre-processing

For each cDNA and pDNA replicate of each library, barcodes were extracted from single-end reads using a custom Python script that matches the constant region after the barcode. Barcodes were clustered using Starcode (version 1.1) (*Zorita et al., 2015*) to remove errors from sequencing and counts were summarised.

## cDNA and pDNA data post-processing

For each library cDNA and pDNA barcodes were matched to iPCR barcodes. Any barcode assigned to multiple fragment identities was removed from the data. Per cDNA/pDNA replicate barcode counts were normalised to the total number of barcode counts. Then, activity was calculated per barcode and cDNA replicate as the cDNA:pDNA normalised counts ratio. Per individual duplet combination these normalised activity ratios were averaged across barcodes with a minimum requirement of five barcodes and eight pDNA counts per barcode. The mean activity across the three replicates was calculated as the geometric mean.

## Calculation of boost indices

In our previous study (*Martinez-Ara et al., 2022*) we observed that it was critical to use control sequences to estimate the baseline activity of the promoters. For each promoter we used all of the CC combinations to estimate its baseline activity. We calculated this as the median activity across all CC combinations. Then, the boost index of an EE or EC (single enhancer boost index) combination was calculated as the $\log_2$ ratio between the observed activity of this combination and the baseline activity of the promoter. Except in *Figure 2—figure supplement 2*, boost indices were averaged across positions and orientations. In *Figure 2a* boost indices were averaged across all EC combinations for each enhancer. In *Figure 3a* boost indices were averaged across orientations but not positions.

## Selectivity of promoters

To analyse the selectivity of promoters on single enhancers (enhancer-control-promoter combinations) we used a Welch F-test due to the heteroscedasticity of the data. The test was performed on each of the populations of enhancer-control-promoter combinations. First the data was confirmed to be approximately normal by a Shapiro test, which confirmed that 94.1% of the 439 single enhancers+promoter combinations (that could be tested) show normal distributions (at a 1% FDR).

Definition of selectivity: We consider selectivity of enhancers and promoters as the differences that lead to an enhancer quantitatively activating some promoters more strongly than others. We prefer the use of selectivity rather than compatibility because compatibility in a biochemical sense is more of an on/off binary descriptor. Therefore, we consider selectivity is a better descriptor of a quantitative difference.

## Calculation of additive and multiplicative expected activities

To calculate the expected additive and multiplicative activities we first had to determine three values. The baseline activity of the promoter ($A_{CCP}$) was measured as the median activity across all CC combinations for each promoter. The single enhancer activity estimates ($A_{E1CP}$ and $A_{CE2P}$, enhancer in position 1 and position 2, respectively) were calculated as the median activity of all enhancer-control combinations where the enhancer was found in either position 1 ($A_{E1CP}$) or position 2 ($A_{CE2P}$). Therefore,

$$A_{CCP} = median\{A_{C1C2P}, A_{C1C3P}, ..., A_{CiCjP}\}$$
$$A_{E1CP} = median\{A_{E1C1P}, A_{E1C2P}, ..., A_{E1CiP}\}$$
$$A_{CE2P} = median\{A_{C1E2P}, A_{C2E2P}, ..., A_{CiE2P}\}$$

Then, for each enhancer-enhancer-promoter combination the expected additive activity was calculated as the sum of the estimated activities of the single enhancers in each position minus the promoter baseline activity ($A_{E1CP} + A_{CE2P} - A_{CCP}$). The multiplicative expected activity was calculated as the product of the estimated activities of the single enhancers in each position divided by the promoter baseline activity (($A_{E1CP} * A_{CE2P}$)/$A_{CCP}$). For the expected additive activity, the SD of the estimate was propagated as the square root of the sum of the variances of the single measurements. Therefore,

$$Expected\ Additive\ Activity = (A_{E1CP} + A_{CE2P} - A_{CCP})$$
$$Expected\ Multiplicative\ Activity = (A_{E1CP} * A_{CE2P})/A_{CCP}$$

The additive expected activities were then used to calculate $log_2$(observed/expected) ratios in order to find supra-additive behaviours between enhancers. To find trends the $log_2$ ratios were then averaged for each enhancer across EE pairs.

## Non-linear promoter responses

For every EE combination present in all eight libraries we averaged boost indices across the eight promoters. This average enhancer effect was then used to calculate the slope between the average boost index and the observed boost index for each of the promoters. The slopes were calculated using the lm function in R.

Definition of non-linear: We consider non-linear any relationship between two variables in the linear space where a change of one variable does not correspond to a proportional change in the other variable that follows a linear equation. Therefore, we consider proportional changes in the log-space as non-linear as they arise from non-linear relationships in the linear space.

## Materials availability

Libraries generated in this study are available upon request.

## Acknowledgements

We thank the NKI Genomics, and Research High Performing Computing facilities for technical support, and members of our laboratory for helpful discussions and comments. Funded by the European Union (European Research Council, RE_LOCATE, 101054449). Views and opinions expressed are however those of the authors only and do not necessarily reflect those of the European Union or the European Research Council. Neither the European Union nor the granting authority can be held responsible for them. Research at the Netherlands Cancer Institute is supported by an institutional grant of the Dutch Cancer Society and of the Dutch Ministry of Health, Welfare and Sport. The Oncode Institute is partially funded by the Dutch Cancer Society.

## Additional information

### Competing interests

Federico Comoglio: co-founder of enGene Statistics GmbH. The other authors declare that no competing interests exist.

### Funding

| Funder | Grant reference number | Author |
|---|---|---|
| European Research Council | 101054449 | Miguel Martinez-Ara Bas van Steensel |

The funders had no role in study design, data collection and interpretation, or the decision to submit the work for publication.

### Author contributions

Miguel Martinez-Ara, Conceptualization, Software, Formal analysis, Investigation, Visualization, Methodology, Writing - original draft, Writing - review and editing; Federico Comoglio, Conceptualization, Formal analysis, Methodology; Bas van Steensel, Conceptualization, Supervision, Funding acquisition, Writing - review and editing

### Author ORCIDs

Miguel Martinez-Ara (iD) http://orcid.org/0000-0002-8219-0754
Bas van Steensel (iD) https://orcid.org/0000-0002-0284-0404

Reviewer #1 (Public review): https://doi.org/10.7554/eLife.91994.3.sa1
Reviewer #2 (Public review): https://doi.org/10.7554/eLife.91994.3.sa2
Author response https://doi.org/10.7554/eLife.91994.3.sa3

---

## Additional files

### Supplementary files

• Supplementary file 1. mm10 genomic coordinates of the regulatory elements tested in this study.

• Supplementary file 2. Sequences of the synthetic controls used in this study.

• Supplementary file 3. Processed and normalised data for all the enhancer-enhancer-promoter (EEP) combinations tested in this study.

• Supplementary file 4. Sequences of the primers used in this study.

• MDAR checklist

### Data availability

Raw sequencing data and processed data are available at Gene expression Omnibus (GEO): GSE240586. Publicly available datasets that were used are listed in and are available at https://osf.io/yuc4e/. Lab journal records are also available at https://osf.io/yuc4e/. Code of data processing pipelines and analysis scripts are available at https://github.com/vansteensellab/EEPCombinations (copy archived at *Martinez-Ara, 2024*).

The following datasets were generated:

| Author(s) | Year | Dataset title | Dataset URL | Database and Identifier |
|---|---|---|---|---|
| Martinez-Ara M, Comoglio F, van Steensel B | 2023 | Large-scale analysis of the integration of enhancer-enhancer signals by promoters | https://www.ncbi.nlm.nih.gov/geo/query/acc.cgi?acc=GSE240586 | NCBI Gene Expression Omnibus, GSE240586 |
| Martinez-Ara M, van Steensel B | 2024 | Large-scale analysis of the integration of enhancer-enhancer signals by promoters | https://doi.org/10.17605/OSF.IO/YUC4E | Open Science Framework, 10.17605/OSF.IO/YUC4E |

The following previously published datasets were used:

| Author(s) | Year | Dataset title | Dataset URL | Database and Identifier |
|-----------|------|---------------|-------------|------------------------|
| Joshi O, Wang SY, Atlasi Y, Peng T, Saeed S, Handoko L, Kuznetsova T | 2015 | DNaseI_2i | https://www.ncbi.nlm.nih.gov/geo/query/acc.cgi?acc=GSM1856456 | NCBI Gene Expression Omnibus, GSM1856456 |
| Joshi O, Wang SY, Atlasi Y, Peng T, Saeed S, Handoko L, Kuznetsova T | 2015 | DNaseI_serum | https://www.ncbi.nlm.nih.gov/geo/query/acc.cgi?acc=GSM1856455 | NCBI Gene Expression Omnibus, GSM1856455 |

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
