## [Editor Report · eLife assessment]

Understanding how genomic regulatory elements control spatiotemporal gene expression is essential for explaining cell type diversification, function, and the impact of genetic variation on disease. This **important** study provides **solid** evidence that enhancers generally combine additively to influence gene expression. Moreover, promoters, particularly weaker ones, can exhibit supra-additivity when integrating enhancer effects. These findings highlight the context-dependent nature of enhancer-promoter interactions in gene regulation, and contribute to ongoing discussions about the selectivity and combination of regulatory elements.

---

## [Referee Report · Reviewer #1 (Public review)]

This manuscript by Martinez-Ara et al investigates how combinations of cis-regulatory elements combine to influence gene expression. Using a clever iteration on massively parallel reporter assays (MPRAs), the authors measure the combinatorial effects of pairs of enhancers on specific promoters. Specifically, they assayed the activity of 59x59 different enhancer-enhancer (E-E) combinations on 8 different promoters in mouse embryonic stem cells. The main claims of the paper are that E-E pairs combine nearly additively, and that supra-additive E-E pairs are rare and often promoter-dependent. The data in this study do generally support these claims.

This paper makes a good contribution to the ongoing discussions about the selectivity of gene regulatory elements. Recent works, such as those by Martinez-Ara et al. and Burgman et al., have indicated limited selectivity between E-P pairs on plasmid-based assays; this paper adds another layer to that by suggesting a similar lack of selectivity between E-E pairs.

An interesting result in this manuscript is the observation that weak promoters allow more supra-additive E-E interactions than strong promoters (Figure 4b). This nonlinear promoter response to enhancers aligns with the model previously proposed in Hong et al. (from my own group), which posited that core promoter activities are nonlinearly scaled by the genomic environment, and that (similar to the trend observed in Figure 5b) the steepness of the scaling is negatively correlated with promoter strength.

My only suggestion for the authors is that they include more plots showing how much the intrinsic strengths of the promoters and enhancers they are working with explain the trends in their data.

Specific Suggestions

Supplementary Figure 4 is presented as evidence for selectivity between single enhancers and promoters. Could the authors inspect the relationship between enhancer/promoter strength and this selectivity? Generating plots similar to Figure 4B and Figure 5B, but for single enhancers, should show if the ability of an enhancer to boost a promoter is inversely correlated to that promoter's intrinsic strength. Also, in Supplementary Figure 4, coloring each point by promoter type would clarify if certain promoters (the weak ones) consistently show higher boost indices across all enhancers. If they do not, the authors may want to speculate how single enhancers can show selectivity for promoters while the effect of adding a second enhancer to an existing E-P has little selectivity. An alternate explanation, based solely on the strength of the elements, would be that when the expression of a gene is low the addition of enhancer(s) have large effects, but when the expression of a gene is high (closer to saturation) the addition of enhancer(s) have small effects.

Can anything more be said about the enhancers in E-E-P combinations that exhibit supra-additivity? Specifically, it would be interesting to know if certain enhancers, e.g. strong enhancers or enhancers with certain motifs, are more likely to show supra-additivity with a given promoter.

Comments on revised version:

The revised manuscript satisfactorily addresses the points I raised in the review. With the addition of the new graphs there is enough data for readers to decide whether the supra-additivity depends only on the strength of the promoter or on some other (undefined) feature of E-P pairs. This manuscript is a solid contribution to the ongoing debate about enhancer-promoter selectivity.

---

## [Referee Report · Reviewer #2 (Public review)]

Summary

This work investigates how multiple DNA elements combine to regulate gene expression. The authors use an episomal reporter assay which measures the transcriptional output of the reporter under the regulation of an enhancer-enhancer-promoter triple. The authors test all combinations of 8 promoters and 59 enhancers in this assay. There are two main findings: (1) enhancer pairs generally combine additively on reporter output (2) the extent to which enhancers increase reporter output over the promoter (individually and as enhancer-enhancer pairs) is inversely related to the intrinsic strength of the promoter. Both of these findings are interesting and are well supported by the data.

This study extends previous results on enhancer-promoter combinations to enhancer-enhancer-promoter triples. For example the near equivalence of Fig. 5b and Fig. S7b is intriguing. This experimental design also provides the ability to investigate the notion of selectivity (also commonly referred to as compatibility) between enhancer-enhancer pairs and promoters.

The authors note many limitations, including the selection of the elements and the size and spacing of the tested elements. Some of the enhancer-enhancer-promoter triples they test were also investigated by a different experimental design in Brosh et al 2023. Brosh et al observed non-additivity between these elements while this study did not. Ultimately we do not know which mechanisms produce the non-additivity that has been observed in native loci and which experimental designs would preserve such mechanisms.

Overall this is a nice experimental design and a great dataset for probing how enhancers and promoters combine to regulate gene expression. I have no major concerns, but I will try to clarify some methodological points I found confusing.

Methodology

The following two comments are meant to help the reader understand the methodology/terminology used in this paper and how it relates to other similar studies.

The interpretation that "promoters scale enhancer signals in a non-linear manner" is potentially confusing. I believe that the authors use "non-linear" to refer to the slopes (represented by the letter 'b' in Fig. 5b) being not equal to 1. Given how the boost index is defined, this implies the relationship

Activity of EEP = (Activity of CCP) * (Average Linear Boost)^b

One potential source of confusion is that the Average Linear Boost term itself depends on the set of promoters that are assayed. Averaging across (many) promoters may alleviate this concern, in which case Average Linear Boost may be considered some form of intrinsic enhancer strength. If so, there is a correspondence between this terminology and the terminology presented in Bergman et al 2022. If b not equal to 1 refers to a non-linear scaling, then the reader may think that b=1 refers to a linear scaling. But if b=1, and the Average Linear Boost term is interpreted as intrinsic enhancer strength, then the equation above implies that the activity of EEP is equal to an intrinsic promoter strength times an intrinsic enhancer strength. This is essentially the relationship that is considered in Bergman et al 2022 and which is referred to in that paper as 'multiplicative'. The purpose of this comment is not to argue for what is the relationship that best explains the data, it is just to clarify the terminology.

Enhancer-promoter selectivity: As a follow-up to a previous study (Martinez-Ara et al, Molecular Cell 2022) the authors mention that the data in this study also shows that enhancers show selectivity for certain promoters. I found the methodology hard to follow, so this section of the review is meant to guide the reader in understanding how the authors define 'selectivity'. The authors consider an enhancer to be not selective if its 'boost index' is the same across a set of promoters. 'Boost index' is defined to be the ratio of the reporter output with the enhancer and promoter divided by the reporter output with just the promoter. Conceptually, I think that considering the boost index is a reasonable way to quantify selectivity. The authors use a frequentist approach to classify each enhancer as selective or not selective. The null hypothesis is that the boost index of the enhancer is equal across a set of promoters. This can be visualized in Fig. 2C where the null hypothesis is that the mean of each vertical distribution is equal. Note that in Figure S4b of this paper (and in Figure 4B of their 2022 paper) the within-group variance is not plotted. Statistical significance is assessed using a Welch F-test.

---

## [Author Response]

The following is the authors’ response to the original reviews.

**Public Reviews**:
**Reviewer #1 (Public Review):**

We thank the reviewer for the positive and constructive comments. We apologize for the very long delay in submitting this revised manuscript; due to personal circumstances we were not able to do this earlier.

This manuscript by Martinez-Ara et al investigates how combinations of cis-regulatory elements combine to influence gene expression. Using a clever iteration on massively parallel reporter assays (MPRAs), the authors measure the combinatorial effects of pairs of enhancers on specific promoters. Specifically, they assayed the activity of 59x59 different enhancer-enhancer (E-E) combinations on 8 different promoters in mouse embryonic stem cells. The main claims of the paper are that E-E pairs combine nearly additively, and that supra-additive E-E pairs are rare and often promoter-dependent. The data in this study generally support these claims.This paper makes a good contribution to the ongoing discussions about the selectivity of gene regulatory elements. Recent works, such as those by Martinez-Ara et al. and Burgman et al., have indicated limited selectivity between E-P pairs on plasmid-based assays; this paper adds another layer to that by suggesting a similar lack of selectivity between E-E pairs.An interesting result in this manuscript is the observation that weak promoters allow more supra-additive E-E interactions than strong promoters (Figure 4b). This nonlinear promoter response to enhancers aligns with the model previously proposed in Hong et al. (from my own group), which posited that core promoter activities are nonlinearly scaled by the genomic environment, and that (similar to the trend observed in Figure 5b) the steepness of the scaling is negatively correlated with promoter strength.

We now discuss the parallel with the Hong 2022 study (Discussion, lines 307-310).

My only suggestion for the authors is that they include more plots showing how much the intrinsic strengths of the promoters and enhancers they are working with explain the trends in their data.

Agreed, see below.

Specific SuggestionsSupplementary Figure 4 is presented as evidence for selectivity between single enhancers and promoters. Could the authors inspect the relationship between enhancer/promoter strength and this selectivity? Generating plots similar to Figure 4B and Figure 5B, but for single enhancers, should show if the ability of an enhancer to boost a promoter is inversely correlated to that promoter's intrinsic strength...

Thank you for the suggestion, we have now repeated the analysis of Figure 5 for EP pairs instead of EEP triplets, and included it as new Supplementary Figure S7. Despite the lower statistical power, the trends are very similar.

...Also, in Supplementary Figure 4, coloring each point by promoter type would clarify if certain promoters (the weak ones) consistently show higher boost indices across all enhancers. If they do not, the authors may want to speculate how single enhancers can show selectivity for promoters while the effect of adding a second enhancer to an existing E-P has little selectivity. An alternate explanation, based solely on the strength of the elements, would be that when the expression of a gene is low the addition of enhancer(s) has large effects, but when the expression of a gene is high (closer to saturation) the addition of enhancer(s) have small effects.

We now added colour coding for each of the promoters in figure S4. We agree this clarifies the contribution of each promoter to the selectivity of each enhancer and it further confirms the responsiveness trends observed in Figure 5.

Can anything more be said about the enhancers in E-E-P combinations that exhibit supra-additivity? Specifically, it would be interesting to know if certain enhancers, e.g. strong enhancers or enhancers with certain motifs, are more likely to show supra-additivity with a given promoter.

Unfortunately, even with the number of enhancers that we tested, we lack statistical power to identify sequence motifs that may favour supra-additivity.

**Reviewer #2 (Public Review):**

We thank the reviewer for the supportive and constructive comments. We apologize for the very long delay in submitting this revised manuscript; due to personal circumstances we were not able to do this earlier.

SummaryThis work investigates how multiple regulatory elements combine to regulate gene expression. The authors use an episomal reporter assay which measures the transcriptional output of the reporter under the regulation of an enhancer-enhancer-promoter triple. The authors test all combinations of 8 promoters and 59 enhancers in this assay. The main finding is that enhancer pairs generally combine additively on reporter output. The authors also find that the extent to which enhancers increase reporter output is inversely related to the intrinsic strength of the promoter.This manuscript presents a compact experiment that investigates an important open question in gene regulation. The results and data will be of interest to researchers studying enhancers. Given that my expertise is in modeling and computation, I will take the experimental results at face value and focus my review on the interpretation of the results and the computational methodology. I find the result of additivity between enhancers to be well supported. The findings on differential responsiveness between promoters are very interesting but the interpretation of such responses as 'non-linear' or 'following a power-law' may be misleading. More broadly, I think a more rigorous description of the mathematical methodology would increase the clarity and accessibility of this manuscript. A major unanswered question is whether the findings in this study apply to enhancers in their native genomic context. Regardless, investigating such questions in an episomal reporter assay is valuable.Main commentsApplicability to native genomic context: The applicability of the results in this paper to enhancers in their native genomic context is unclear. As the authors state in the discussion section, the reporter gene is not integrated into the genome, the spacing between enhancers does not match their native context etc. It is thus unclear whether this experimental design is able to detect the non-additivity between enhancers which is known to be present in the genome. This could be investigated by testing the enhancer-enhancer-promoter tuples for which non-additivity has been observed in the genome (references are given in the introduction) in this assay.

We appreciate the suggestion, but we chose not to go back to the lab to generate additional data to address this point. Of the cited previous studies, two are comparable to our study because they also used mESCs and included loci that we also studied: Thomas et al. (2021) and Brosh et al. (2023). We now discuss how the findings of these two studies relate to our observations in the Discussion, lines 336-345.

Interpretation of promoter responses as non-linear and following a power-law: In Fig 5, the authors demonstrate that enhancer-enhancer pairs boost reporter output more for weak promoters as opposed to strong promoters. I agree the data supports this finding, but I find the interpretation of such data as promoters scaling enhancers according to a power-law (as stated in the abstract) to be misleading. As mentioned on line 297, it is not possible to define an intrinsic measure of enhancer strength, thus the authors assign the base of the power-law to be the average boost index of the enhancer-enhancer pair across the 8 promoters. But this measure incorporates some aspect of a promoter and is not solely a property of enhancers...

We agree that the power-law conclusion in the abstract was too strong; we have rephrased it as "non-linear".

...It would also be useful to know whether the results in Fig 5 apply to only enhancer-enhancer-promoter triples or also to enhancer-promoter pairs.

We have now added this EP analysis as new Supplemental Figure S7. Although the statistical power is much lower, this shows very similar trends as the EEP analysis. We briefly report this, lines 275-278.

Enhancer-promoter selectivity: As a follow-up to a previous study (Martinez-Ara et al, Molecular Cell 2022) the authors mention that the data in this study also shows that enhancers show selectivity for certain promoters. The authors mention that both studies use the same statistical methodology and the data in this study is consistent with the data from the 2022 paper. However, I think the statistical methodology in both studies needs further exposition. This section of the review is thus meant to ensure that I understand the author's methodology, to guide the reader in understanding how the authors define 'selectivity', and to probe certain assumptions underlying the methodology.My understanding of the approach is as follows: The authors consider an enhancer to be not selective if its 'boost index' is the same across a set of promoters. 'Boost index' is defined to be the ratio of the reporter output with the enhancer and promoter divided by the reporter output with just the promoter. Conceptually, I think that considering the boost index is a reasonable way to quantify selectivity.The authors use a frequentist approach to classify each enhancer as selective or not selective. The null hypothesis is that the boost index of the enhancer is equal across a set of promoters. This can be visualized in Fig. 2C where the null hypothesis is that the mean of each vertical distribution is equal. Note that in Figure S4 of this paper (and in Figure 4B of their 2022 paper) the within-group variance is not plotted. Statistical significance is assessed using a Welch F-test. This is a parametric test that assumes that the observations within each vertical distribution in Fig 2C are normally distributed (this test does allow for heteroskedasticity - which means that the variance may differ within each vertical distribution). Does the normality assumption hold? This analysis should be reported. If this assumption does not hold, is the Welch test well calibrated?

We have tested the normality of all of the single enhancer + promoter combinations that were tested using the welch F-test. 94.1% of the 439 single enhancers + Promoter combinations show normal distributions (at a 1% FDR). We have added this to the methods section of the revised manuscript. Apart from this, non-normality has little to no influence on the Welch F-test performance (https://rips-irsp.com/articles/10.5334/irsp.198). Therefore, the use of the Welch F-test to score enhancer selectivity on these data is valid. Apart from this, we agree that a simple binary classification of selective vs non-selective is not descriptive enough for these kinds of data. We addressed this in our previous publication by exploring the relationship between selectivity and enhancer strength. However, in the objective in this publication was solely to show that this new dataset follows similar selectivity patterns to our previous publication. Furthermore, our analysis on the non-linearity of promoter response is a more quantitative continuation on the analysis on selectivity as this is probably one of the major contributors to enhancer selectivity. This was probably present in our previous paper but could not be analyzed as there were less combinations per promoter.

For further clarity, we have now highlighted the individual promoters in Figure S4 by colors.

**Recommendations for the authors:**

**Reviewer #2 (Recommendations for the authors):**
I found this to be an interesting manuscript and am glad this experiment was conducted. As I wrote in my public review, I think that clarifying the computational methods/ideas would really help. I also think it would be helpful to properly define the terms that are being used. For example, this manuscript uses the terminology cooperativity and synergy. Are these meant to be synonymous with supra-addivity?

Thank you for this point. The revised manuscript no longer uses the word “cooperativity”. We now use “supra-additivity” when describing our data, and “synergy” as biological interpretation. In the Introduction we now clarify this distinction.

Comments on enhancer selectivity:In the public review, I have given comments on the statistical methodology employed to assess enhancer selectivity. On a more subjective note, I'm not convinced that a frequentist approach to a binary classification of 'selective' vs 'not selective' is that useful here. I think it would be more useful to report an 'effect size' of the extent to which an enhancer is selective and to study the sources of this effect size. I think you've tried to do this in lines 329-339 of the discussion but I think the exposition could be clearer.Figure S4B may suggest how to do this. It appears that the distribution of boost indices for a given enhancer is trimodal (this is most obvious for the stronger enhancers on the top of the plot). Is it the case that each mode (for each enhancer) consists of the same set of promoters? I think what is implied by Figure 5B is that the stronger promoters are not boosted as much as the weaker promoters. So does the leftmost mode consist of Ap1m1, the middle mode consist of Klf2/Otx2/Nanog, and the rightmost mode of Sox2/Fgf5/Lefty1/Tbx3? If so, I would recommend emphasizing this in the text/figure and clarifying how this relates to selectivity. It seems that the chain of logic is as follows: (1) We define an enhancer to be selective if its boost indices across a set of promoters are not the same. (2) We generally observe that stronger promoters get boosted less than weaker promoters. (3) Thus selectivity arises due to differences in intrinsic strengths of the promoter. I think this is what is being implied in lines 329-339 of the discussion, but it took me multiple readings to understand this and I'm not convinced the power-law explanation is justified (see public review).

We have modified this paragraph of the Discussion (now lines 350-359).

Regarding the power-law: in the Results we state “*roughly* a power-law function”. We have removed the power-law claim from the abstract, that conclusion as phrased was indeed too firm.

Reference to Zuin et alLines 323 - 325: A reference is made to the data from Zuin et al "following approximately a power-law". What data in Zuin et al does this statement refer to? I do not believe the authors in Zuin et al claim that the relationship between GFP intensity and enhancer-promoter distance (Figure 1h,i from Zuin et al) follows a power law. It is certainly non-linear, but I have taken a look at this data myself and do not find it follows a power-law. Please either explain this further and rigorously justify the claim or adjust the wording accordingly.

Good point, in the discussion of Zuin et al we have replaced “power law” with “non-linear decay function”